# Transcriptome Profile of the Rat Adrenal Gland: Parenchymal and Interstitial Cells

**DOI:** 10.3390/ijms24119159

**Published:** 2023-05-23

**Authors:** Marianna Tyczewska, Patrycja Sujka-Kordowska, Marta Szyszka, Karol Jopek, Małgorzata Blatkiewicz, Ludwik K. Malendowicz, Marcin Rucinski

**Affiliations:** Department of Histology and Embryology, Poznan University of Medical Sciences, Swiecickiego 6 Street, 60-781 Poznan, Poland; psujka@ump.edu.pl (P.S.-K.); mszyszka@ump.edu.pl (M.S.); kjopek@ump.edu.pl (K.J.); mblatkiewicz@ump.edu.pl (M.B.); lkm@ump.edu.pl (L.K.M.); mrucinski@ump.edu.pl (M.R.)

**Keywords:** rat, adrenal gland, transcriptome, global gene profiling, functional annotation clustering, parenchymal cells, interstitial cells

## Abstract

The homeostasis of the adrenal gland plays a decisive role in its proper functioning, both in non-stressful conditions and under the influence of various types of stress. This consists of interactions between all types of cells that make up the organ, including parenchymal and interstitial cells. The amount of available information on this subject in the rat adrenal glands under non-stressful conditions is insufficient; the aim of the research was to determine the expression of marker genes for rat adrenal cells depending on their location. The material for the study consisted of adrenal glands taken from intact adult male rats that were separated into appropriate zones. Transcriptome analysis by means of Affymetrix^®^ Rat Gene 2.1 ST Array was used in the study, followed by real-time PCR validation. Expression analysis of interstitial cell marker genes revealed both the amount of expression of these genes and the zone in which they were expressed. The expression of marker genes for fibroblasts was particularly high in the cells of the ZG zone, while the highest expression of specific macrophage genes was observed in the adrenal medulla. The results of this study, especially with regard to interstitial cells, provide a so far undescribed model of marker gene expression of various cells, both in the cortex and medulla of the sexually mature rat adrenal gland. The interdependence between parenchymal and interstitial cells creates a specific microenvironment that is highly heterogeneous within the gland with respect to some of the interstitial cells. This phenomenon most likely depends on the interaction with the differentiated parenchymal cells of the cortex, as well as the medulla of the gland.

## 1. Introduction

The history of discovering the structure, development, and function of the adrenal gland was described in detail this year by Miller and White [1]. The gland is comprised of two distinct parts: the cortex and the medulla. The adrenal cortex originates from the mesoderm, while the medulla is derived from the neuroectoderm. The adrenal cortex is an integral part of the hypothalamic-pituitary-adrenal axis (HPA), while the medulla is a component of the sympatho-adrenomedullary system.

In these systems, the adrenal cortex synthesizes and secretes a variety of steroid hormones, while the medulla of the gland synthesizes and secretes catecholamines. Pioneering studies by Wurtman’s team [2,3] demonstrated that the glucocorticoids of the rat adrenal cortex regulate the secretory function of the adrenal medulla. Over the years, the functional interrelationships between the cortex and medulla of the gland have been more thoroughly understood. A number of systems involved in this regulation have been recognized, and their mechanisms of action have been clarified. The presence of different cell types in the gland (adrenocortical, chromaffin, endothelial, immune, etc.) forms the basis of an interplay termed crosstalk. This interaction is mediated most often via biologically active peptides and catecholamines, mainly through their direct action on adjacent cells or in a paracrine manner. In this way, a specific microenvironment is generated throughout the adrenal gland that regulates the expression levels of various genes. These phenomena are described in detail in a series of elegant articles, which are mostly reviews [4,5,6,7,8,9,10].

The aforementioned mutual interactions between all types of adrenal gland cells maintain homeostasis in the adrenal gland, both under physiological and pathological conditions. This homeostasis is reflected in the gene expression levels of both the cortex and medulla of the gland. In this aspect, there are numerous publications on the human adrenal gland under physiological and pathological conditions, mainly in tumors of the cortex and medulla of the gland [11,12,13,14,15,16,17,18,19,20]. These studies used a wide range of modern methods, including QPCR, NGS and single-cell sequencing, which allow precise recognition of the transcriptome in individual components of the gland.

Much less information is available on the precise analysis of the transcriptome in the rat adrenal gland. The transcriptional profile of genes in the rat adrenal cortex is relatively well characterized [21,22]. In contrast, much less information is available on the gene expression profile in the rat adrenal medulla. Studies of this type are mainly concerned with the transcriptome profile of adrenal medulla genes under different types of stress. Moreover, it is rare that this expression was determined in the isolated medulla of the gland. Because the adrenal glands of females are more susceptible to stress, they are mainly carried out on female rats [23,24,25,26,27].

In the literature available to us, we found no reports of gene expression profile studies in the rat adrenal gland that involved both parenchymal and interstitial cells in non-stressful conditions. Therefore, the aim of this study was to analyze the gene expression levels in different types of rat adrenal building cells, both parenchymal and interstitial cells, depending on their location. The results of such analyses can provide a basic model to describe the adrenal transcriptome of the rat under basal conditions (intact rats). To avoid changing estrogen levels during the estrous cycle, the study was performed on adult male rats [22,28].

## 2. Results

The results of the microarray analysis were analyzed using principal component analysis (PCA) (Figure 1). This method demonstrated that gene combinations can distinguish between different clusters corresponding to the various zones of the adrenal gland, including the glomerulosa (ZG) and fasciculate/reticularis (ZF/R) zones, and the medulla (M). As shown in Figure 1A, the genes of the different zones exhibit clear separation from each other. The PCA graph shows that the ZG and ZF/R zone gene groups have dispersed in the left part of the graph, separating from the adrenal M gene groups, which have dispersed in the right part of the graph (Figure 1A). Correlation analyses between samples show high homogeneity and confirm the similarity between adrenal cortex compartments, expressed by a high correlation coefficient (R) between ZG and ZF/R (R range 0.37–0.86) (Figure 1B).

Using the Affymetrix microarray method, we carried out expression profiling of genes derived from individual adrenal zones of the rat. The same cut-off criteria were used for all comparisons (abs(fold change) > 2 and *p* < 0.05 with 20% FDR correction). Overall expression profiles are presented as volcano plots (Figure 2A). All study groups were compared with each other (ZG vs. ZF/R, ZG vs. M, ZF/R vs. M). When comparing ZG and ZF/R zones, only 53 genes were down-regulated, and as many as 394 were up-regulated. Considering the cut-off criteria, the largest changes in gene expression were observed between the cortical zones and the adrenal medulla. Thus, in the ZG vs. M comparison, 712 genes showed a decrease in expression and 393 increase, while in the ZF/R vs. M comparison, as many as 870 genes were down-regulated and 209 up-regulated. The list of genes with the greatest changes in expression (up or down) is shown in part B of Figure 2. Amongst the ZG vs. ZF/R genes, the following genes showed a significant increase in expression (about 20 times): angiotensin II receptor, type 2 (Agtr2, fold change: 27.60); angiotensin II receptor, type 1b (Agtr1b, fold change: 18.08); lumican (Lum, fold change: 23.15); visinin-like 1 (Vsnl1, fold change: 23.94); leucine zipper protein 2 (Luzp2, fold change: 23.96); or cytochrome P450, family 11, subfamily b, polypeptide 2 (aldosterone synthase, Cyp11b2, fold change: 17.53). 

Out of the genes with the greatest decrease in expression were the ABI family member 3 binding protein (Abi3bp), potassium voltage-gated channel subfamily C member 2 (Kcnc2), vannin 3 (Vnn3), malic enzyme 3 or hepcidin antimicrobial peptide (Hamp). Within the ZG vs. M gene group, the greatest increase in expression was seen among the genes: Cyp11b2, fold change 40.03; Luzp2, fold change 27.12; Vsnl1, fold change 22.92; fatty acyl CoA reductase 2 (Far2), fold change 18.64. The group of genes with the most statistically significantly reduced expression included: solute carrier family 18 member A1 (Slc18a1), fold change (−29.05); solute carrier family 24 member 2 (Slc24a2), fold change (−30.82); GATA binding protein 3 (Gata3), fold change (−24.18); dispatched RND transporter family member 2 (Disp2), fold change (−29.98); and synaptosome associated protein 25 (Snap25), fold change (−33.40). In the most recent comparison, ZF/R vs. M, the genes with significantly increased expression included those whose expression increased by just over three times. In contrast, genes with significantly reduced expression included those whose expression increased more than 20-fold, e.g.: Slc24a2, Snap25 or neuronatin (Nnat).

In this study, we analyzed the expression patterns of genes in each adrenal zone (see Figure 3). To do this, we extracted expression data only for those genes whose expression was significantly up- or down-regulated in each zone (i.e., genes with the highest or lowest expression in ZG, ZF/R or M). We then performed clustering analysis to group genes based on their expression patterns.

The first cluster included genes with the highest expression in the zona glomerulosa (N = 342) and low expression in the other zones. The second cluster comprised genes with the highest expression in the ZF/R (N = 103) and low expression in ZG and M. The third cluster contained genes with the highest expression exclusively in the adrenal medulla (N = 750). Clusters four to six comprised genes with particularly low expression in the ZG (cluster 4, N = 16), ZF/R (cluster 5, N = 135), and M (cluster 6, N = 72) zones, respectively (see Figure 3). The colors in the graphs correspond to the legend, where purple represents the least change in expression and yellow represents the highest.

Then, we presented the expression profiles of the most significantly changed genes from the clusters in Figure 3 using a heatmap (Figure 4). The left panel shows the expression profile of clusters one and four genes, whose expression is either high in the ZG zone (top) or low (bottom). In the other zones, the expression of the genes studied is clearly opposite. Among the up-regulated genes in the ZG zone were fatty acyl CoA reductase 2 (Far2), a regulator of G-protein signaling 6 (Rgs6), G protein subunit alpha 14 (Gna14), somatostatin receptor 2 (Sstr2), Lum, Vsnl1, immunoglobulin superfamily, member 1 (Igsf1), nuclear receptor subfamily 0, group B, member 1 (Nr0b1), Luzp2. Among the genes with lower expression are: Kcnc2, SPARC-related modular calcium binding 1 (Smoc1), alpha-2-macroglobulin (A2m), Vnn3 and malic enzyme 3 (Me3). In the middle panel, the genes with the highest (bottom) or the lowest (top) expression in the fasciculate/reticularis zone (clusters 2 and 5) are collected. It can be seen that the expression of the genes presented if it is low in the ZF/R, is higher in the other zones, and if it is high in the ZF/R, it is lower in the other zones, especially in the adrenal medulla. 

The group of genes with low expression in the ZF/R included: Rgs6, dermatopontin (Dpt), mesoderm specific transcript (Mest), glutathione peroxidase 3 (Gpx3), cell adhesion molecule 1 (Cadm1), sodium voltage-gated channel alpha subunit 7 (Scn7a), dipeptidyl peptidase like 6 (Dpp6), Agtr2 (Figure 4). In the group of genes with increased expression are fibroblast growth factor receptor 2 (Fgfr2), cytochrome P450, family 1, subfamily b, polypeptide 1 (Cyp1b1), neuronal cell adhesion molecule (Nrcam), hemopexin (Hpx), prostaglandin reductase 1 (Ptgr1), neuronal pentraxin 2 (Nptx2). In the right-hand panel, genes whose expression was particularly markedly down-regulated (bottom) or up-regulated (top) in the adrenal medulla alone (clusters three and six) are included. The graph shows that the expression profiles and genes with elevated expression in the medulla include carbonic anhydrase 12 (Ca12), C-C motif chemokine ligand 11 (Ccl11), Ras-like without CAAX 2 (Rit2), GATA binding protein 3 (Gata3), neuropeptide Y (Npy), dispatched RND transporter family member 2 (Disp2), solute carrier family 24 member 2 (Slc24a2), Snap25 and GNAS complex locus (Gnas) and neuronatin (Nnat). In contrast, genes with markedly lower expression include ephrin B2 (Efnb2), RAB20, member RAS oncogene family (Rab20), regucalcin (Rgn), prostaglandin D2 receptor 2 (Ptgdr2) (Figure 4).

The genes studied from the different adrenal zones, as shown in Figure 2, were then subjected to additional enrichment analysis using the DAVID GO BP FAT database, as shown in Figure 5. A list of ontological processes to which the analyzed genes were assigned was obtained. The larger the bubble, the greater the number of genes in the ontological process, and the more transparent the bubble, the more statistically significant (closer to *p*-value) the changes in expression. An increase or decrease in the expression of genes involved in specific GO processes was marked with the corresponding color, green if there was an increase in expression and red if the genes were down-regulated. Each zone is shown in a separate rectangle. As shown in the graph, the least ontological processes involve ZG zone genes. Of these, the positive and negative regulation of the canonical Wnt pathway (GO:0090263 and GO:0090090), cellular response to transforming growth factor beta stimulus (GO:0071560), negative regulation of cell migration (GO:0030336), and extracellular matrix organization (GO:0030198) stand out (Figure 5 left). 

Among the ontological processes involving ZF/R zone genes were those whose expression primarily declined. These are collagen fibril organization (GO:0030199), extracellular matrix organization (GO:0030198), cell adhesion (GO:0007155), and cGMP biosynthetic process (GO:0006182). Only the oxidation–reduction process involved ZF/R zone genes whose expression increased. The largest number of ontology groups was obtained for genes whose expression was altered in the adrenal medulla (Figure 5 right). The expression of most increased, as can be seen by the color of the bubbles. Among the ontological groups, therefore, one can distinguish those related to the regulation of the postsynaptic potential (GO:0097151, GO:0086010, GO:0060079), membrane depolarization during the action potential (GO:0086010), the response to various stimuli (GO:0071560, GO:0071377), ion transmembrane transport (GO:0071805, GO:0070588, GO:0034765, GO:0034220), the positive regulation of GTPase activity (GO:0043547), and the biosynthesis of catecholamines (GO:0042423). Furthermore, signal transduction (GO:0007165), chemical synaptic transmission (GO:0007268), behavioral fear response (GO:0001662), nervous system development (GO:0007399, GO:0048485) and neuronal migration (GO:0001764).

The next figure shows the schematic Wnt signaling pathway, in which genes that are significantly regulated in the ZG zone are involved (Figure 6). Each gene relevant to the pathway is marked with a separate rectangle. Changes in the expression of a particular gene (up- or down-regulated) have been highlighted in the corresponding color in the diagram, green if the expression has increased and red if it has decreased. It can be seen that the changes in gene expression were increased, not decreased. Among the up-regulated genes of the signaling pathway analyzed were proteins involved in various steps of the pathway, including Wnt family members 4 and 2b (Wnt4, Wnt2b), secreted frizzled-related proteins (Sfrp1, Sfrp4, Sfrp2), frizzled class receptors (Fzd7, Fzd4), coreceptor Ror 1, phospholipase C beta 1 (Plcb1), NKD inhibitor of WNT signaling pathway 1 (Nkd1), planar cell polarity protein PRICKLE (Prickle1), nuclear factor of activated T-cells 4 (Nfatc4), transcription factor 7 (Tcf7), and cyclins D1 and D3 (Ccnd1, Ccnd3).

We also revealed the pathway of biosynthesis of biogenic amines, in which the majority of genes involved in these signaling pathways were positively stimulated, and their expression was statistically significantly altered, but only in the rat adrenal medulla (Figure 7). Among the genes examined, the expression of which was up-regulated are essential enzymes, such as tyrosine hydroxylase (Th), dopa decarboxylase (Ddc), histidine decarboxylase (Hdc), dopamine beta-hydroxylase (Dbh), phenylethanolamine-N-methyltransferase (Pnmt) and acetylcholinesterase (Ache).

We then carried out an analysis of the assignment of specific genes to specific ontology groups using the cluster profiler library (Figure 8). For the analysis, the genes whose expression was most strongly regulated in each zone of the rat adrenal gland were each selected and assigned to specific ontological processes, up to a maximum of five most regulated GOs, according to the conditions. As shown in the diagram, each zone contains processes that are significantly regulated by a large number of genes. In the glomerular zone of the adrenal cortex, the most highly regulated processes include urogenital development, a process regulated by 32 genes; extracellular matrix organization and extracellular structure organization regulated by more than 40 genes; negative regulation of cellular component movement and negative regulation of locomotion, which is controlled by more than 30 genes. 

In contrast, in the fasciculata/reticularis zones of the cortex, the most highly regulated ontological processes include the carboxylic acid catabolic process, cellular modified amino acid metabolic process, electron transport chain, organic acid catabolic process and small molecule catabolic process. In contrast, in the adrenal medulla, the most strongly regulated ontological processes included: modulation of chemical synaptic transmission, regulation of ion transmembrane transport, regulation of membrane potential, regulation of neurotransmitter levels and the regulation of trans-synaptic signaling (Figure 8).

The next graph shows a comparison of our results obtained in the present study with those of Nishimoto et al. [21] (Figure 9). The scheme shows genes whose expression was statistically significantly altered in the adrenocortical zones, ZG and ZF. The results of the gene expression analysis are shown in circles of different colors, and fold-change is shown according to the legend. Thus, the number of genes and abbreviated names obtained in this study in ZG are shown in pink (320) and ZF in blue (48). The results of the gene expression analysis from the Nishimoto et al. publication [21] are shown in a circle colored grey (ZG: 160 genes) and green (ZF: 226 genes). The number of genes common to both analyses in the ZG zone, in which the expression level was elevated, is 74 genes, including Cyp11b2, delta-like non-canonical Notch ligand 1 (Dlk1), immunoglobulin superfamily, member 1 (Igsf1), Sstr2, Rgs4, Smoc2, although their expression was higher in our study. The number of genes common to both analyses in the ZF zone is only five genes: Cyp1b1, Hpx, Hamp, F11 receptor (F11r) and myomesin 2 (Myom2) (Figure 9). The expression of these genes was at a comparable level. No comparison was made for the adrenal medulla.

We also conducted an analysis of gene expression markers in various adrenal cells. Figure 10 and Figure 11 show the expression profile of marker genes in different types of cells that create the characteristic microenvironment of the gland. Figure 10 presents the marker genes of the adrenal parenchymal cells, i.e., cells of the glomerulosa, fasciculata, and reticularis zones, as well as the medulla. The expression of marker genes in the parenchymal cells of the cortex (cytochrome P450, family 21, subfamily a, polypeptide 1 (Cyp21a1), hydroxy-delta-5-steroid dehydrogenase, 3 beta- and steroid delta-isomerase 2 (Hsd3b), steroidogenic acute regulatory protein (Star), cytochrome P450 family 11 subfamily B member 1 (Cyp11b1), Cyp11b2, cytochrome b5 type A (Cyb5a)) is significantly higher in the corresponding cortical zones compared to the medulla, while the expression of marker genes in adrenal medulla cells (chromogranin A (Chga), phenylethanolamine-N-methyltransferase (Pnmt), tyrosine hydroxylase (Th)) is clearly higher in the medulla of the gland (Figure 10).

On the other hand, Figure 11 shows the expression of marker genes in other types of cells that make up the adrenal microenvironment, such as vascular endothelial cells, fibroblasts, macrophages, lymphocytes, and glial cells. As can be observed, in the case of both vascular endothelial cells, fibroblasts, and smooth muscle cells of blood vessels, characteristic genes that are expressed in specific zones of the gland have been identified. For instance, the genes Cdh5 (cadherin 5) and Tm4sf (transmembrane 4 superfamily) exhibit significantly higher expression in endothelial cells in the medulla and zona glomerulosa of the adrenal cortex, respectively, while the expression of Pecam1 does not show any differences between the gland zones. In turn, the presented marker genes of VSCM (actin alpha 2 (Acta2) and transgelin (Tagln)) are highly expressed in the medulla than in the cortex, except for Myh11 (myosin heavy chain 11), whose expression is comparable in the zona glomerulosa and medulla. Moreover, all the presented fibroblast marker genes (type I collagen: Col1a1, Col1a2, and Lum) are characterized by significantly higher expression in the zona glomerulosa than in other zones of the organ. In turn, the expression of the studied macrophage marker genes is significantly higher in the adrenal medulla compared to the zona glomerulosa and zona fasciculate/reticularis. However, we did not observe significant differences in the expression of lymphocyte genes, which were comparable between the cortex and medulla zones, except for Cd79a molecule. On the other hand, the expression of Cryab (crystallin alpha B) and Plp1 (proteolipid protein 1) in glial cells was higher in the medullary cells (Figure 11).

As a final analysis, we validated the expression of selected genes by real-time qPCR and obtained results that confirmed the microarray analysis (Figure 12). In this analysis, we selected genes that were not markers of specific rat adrenal cells, but their expression levels in the gland were very high. All primers used in the analysis are listed in Table 1. In the case of the Ca12 gene, its expression proved to be the highest and statistically significant in the adrenal medulla. The expression of the remaining genes (Dpt, Lum, Wnt2b) was statistically significantly higher in the glomerular zone of the cortex. Only the expression levels of the Fzl4 gene were similar in all adrenal components studied (ZG, ZF/R and M).

## 3. Discussion

As mentioned in the introduction, cells of the adrenal cortex develop from the mesoderm, while chromaffin cells, which form a distinct medulla in most mammals, are derived from the neuroectoderm. Despite these embryological, structural and functional differences, the two parts of the gland show numerous functional interrelationships. As early as 1966, Wurtman and Axelrod [2] showed that adrenal cortex steroids are involved in the control of adrenaline biosynthesis and secretion. In turn, the catecholamines of the adrenal medulla, most likely through a paracrine pathway, are capable of stimulating adrenal steroid synthesis and secretion [29,30]. Interstitial cells of the gland must also be involved in these interactions, which should be reflected in the gene expression profile both of the cortex and medulla of the gland.

For human adrenal glands, there are numerous publications on transcriptome profiles, both in normal and pathological conditions, especially for adenomas and carcinomas of this organ. Many of these concern both parenchymal and interstitial cells of the gland [11,12,18,31,32,33].

As is well known, in experimental studies, the rat is one of the most common experimental models. In this aspect, there are very numerous publications on gene expression analysis, but they are usually limited to selected metabolic pathways or zones of the cortex or medulla of the adrenal gland. These publications mainly deal with a variety of experiments or developmental aspects of the organ. In contrast, analyses of the rat adrenal transcriptome that consider both cortical (ZG, ZF and ZR) and medullary (M) components under “intact” conditions, including both parenchymal and interstitial cells, are rare. Noteworthy in this aspect is the description by Yu et al. [34] of a transcriptomic BodyMap of male and female rats in aspects of developmental and sex differences, with analysis performed on whole adrenal glands using the RNA-seq method. However, studies of this type do not document gene expression in individual components of the gland.

To date, the most complete gene expression profile in the rat adrenal glands under intact conditions, taking into consideration the localization of these genes, was described by Nishimoto et al. [21]. The authors performed a detailed analysis of the transcriptome in the ZG and ZF zones of intact adult male rats. Nishimoto’s study utilized frozen adrenal sections from Sprague-Dawley rats obtained through laser capture microdissection and stained with Cresyl violet. On the other hand, expression analysis was conducted using the Rat Ref-12 array method. The application of such experimental techniques and the utilization of a distinct rat strain could have influenced the disparities observed between our findings and those of Nishimoto.

In this study, we compared our results to theirs [21] and found some genes with similar expression patterns. As it appeared, in the ZG, a total of 74 genes showed high expression levels in both analyses, including Cyp11b2, Agtr1a, Agtr1b, Dlk1 and Smoc2, with their expression levels found to be higher in our study. However, only five genes were found to be common to both analyses in the ZF, although their expression levels were comparable. The results were not surprising, indicating that the most significant changes in gene expression observed in the studied zones of the adrenal cortex involved marker genes, primarily those encoding steroidogenesis enzymes or associated with its regulation, positive and negative regulation of Wnt signaling pathways [35,36,37,38]. This finding is not surprising, given the vital role these cells play in steroid hormone biosynthesis and secretion. 

On the other hand, several genes showing altered expression have also been implicated in the regulation of cell proliferation and cell growth (Dlk1, Smoc2), intracellular signaling pathways (Rgs4, Rgs6), in the structural remodeling of the extracellular matrix (ECM). Some of them are involved in the regulation of immune response (Serpinb9, Serping1), and others participate in the regulation of interactions between cells (Igsf1) or cell organization and differentiation, inducing intercellular interactions by promoting migration, such as Eph receptors [39]. The differences in the expression levels of some genes observed in the two studies are most likely due to methodological differences. Nishimoto et al. [21] studied ZF cells precisely, while our samples contained portions of ZF and ZR zones, so we refer to them as ZF/R.

In the present study, we also focused on comparing the gene expression profile of different zones of the adrenal cortex with the medulla of the gland, under non-stress conditions (intact adult male rats). Considering the origin of the chromaffin cells of the adrenal medulla, as well as their role in the biosynthesis and secretion of catecholamines and neuropeptides, it is obvious that their transcriptome profile must be significantly different from that of adrenocortical cells. The results of our analysis confirm this.

It appeared that the number of genes with significantly different expression levels was the highest when comparing ZG and ZF/R with M. Specifically, the analysis revealed that 712 genes were down-regulated and 393 were up-regulated in the ZG vs. M components of the gland. Similarly, 870 genes were down-regulated, and 209 were up-regulated in the ZF/R vs. M. Thus, transcriptome analysis allowed us to identify a significant number of genes whose expression was different in the adrenal compartments, although not all genes exhibited high and statistically significant changes in their expression levels. Among the most highly regulated genes were Cyp11b2, Agtr1b, Agtr2, Vsnl1, Lum, Dpt, Nr0b1, Igsf1, Far2 and Sstr2, as well as Caln1, Efnb2, Npy or Gnas, which were either specific to the cortex or the medulla of the adrenal gland. An example of a gene that is strongly up-regulated in medullary cells is Nnat and Snap25, which play a role in forming and maintaining the structure of the nervous system or Ca12, whose significantly up-regulated expression was confirmed by real-time qPCR. It is interesting that according to current knowledge, different Cas can stimulate the steroidogenesis process [8,40]. Some differentially expressed genes belong to a group of transcription factors and nuclear proteins, as well as genes associated with cell signaling, proliferation and cell death. Additionally, genes involved in immune response and metabolism have also been found to be upregulated [25,26,27].

The analysis of ontological processes identification revealed that the medulla of the gland had the highest number of assigned GO processes, and the majority of them contained genes that were up-regulated. These processes were mainly related to postsynaptic potential, catecholamine biosynthesis, neuronal cell migration and stress response. In addition, as our results show, under physiological, non-stressful conditions, the genes with altered expression differ from those activated or inhibited under stressful conditions, as mentioned before [25,26,27].

As we have already mentioned, there is a lack of publications in the literature characterizing the adrenal-specific microenvironment of the intact adult rat adrenal gland, which, mainly through paracrine mechanisms, regulates the functions of both the cortex and medulla of the gland [5,7,8,41,42]. Most of the available data on this topic pertains to interactions occurring under pathological conditions in the human adrenal gland, such as during sepsis or other stressful conditions, as previously described [18,32,40,43].

Along with the parenchymal cells of the adrenal gland, the interstitial cells of the gland play an important role in the formation and maintenance of this microenvironment. They belong to the connective tissue (extracellular matrix formation), vascular, nervous, or immune systems. Unfortunately, in the rat, the number of marker genes of adrenal interstitial cells is limited. That is because many cells, for example, mast cells [44,45] and pericytes [46,47] reveal organ-specific heterogeneity. In our study, we based our research on the marker genes used by Huang [18] with appropriate modifications.

As our study revealed, the expression of fibroblast marker genes, Col1a1, Col1a21 and Lum, is highest in the ZG, while much lower in the remaining parts of the gland. As is well known, fibroblasts are responsible for the formation of ECM (extracellular matrix). In this regard, ECM-derived proteins, including fibronectin, laminin and collagen IV, can either inhibit or stimulate the secretory activity of human fetal adrenal cells in primary culture [48,49]. Moreover, the interaction between ECM factors and adrenal cells may also have an impact on the development of adrenal tumors [50]. Both our study and Nishimoto’s study [21] showed significantly elevated levels of Lum, Dpt, Col1a2, Col6a3 genes expression in the ZG zone, which are known to be associated with the reorganization of ECM [21,51,52]. As previously demonstrated, lumican, a marker of adrenal cortex-localized fibroblasts, plays a significant role in the organization of connective tissue collagen fibers, while Dpt interacts with other enzymes to regulate extracellular matrix formation [18,51].

A specific model of adrenal vascularization of mammals, including the rat, is known. In this gland, there is an interrelationship between blood flow and steroid secretion rates. In addition, both endothelial cells and VSMCs (vascular smooth muscle cells), through a number of secreted active substances, are involved in the regulation of both the cortex and the adrenal medulla [4,53,54,55,56,57,58,59]. We identified elements of this system in the rat adrenal gland using endothelial cell marker genes and VSMCs. The observed expression levels of Cdh5, Pecam1 and Tm4sf1 genes show significant heterogeneity of endothelial cells in the gland. All of these genes show the lowest expression levels in ZF/R. Cdh5 expression is highest in ZG, while Tm4sf1 gene expression is highest in M. This result suggests notable heterogeneity of the endothelial cells in the rat adrenal gland, but the biological significance of this diversity requires further study. In contrast, the expression of VSMC marker genes is lowest in ZF/R, and significantly higher in ZG and M. In this case, by examining the markers of the three genes, we observed very similar levels of their expression in the adrenal compartments studied.

Another group of rat adrenal interstitial cells we analyzed are cells that are components of the immune system. The cells of this system are an important element in creating and maintaining the microenvironment and regulating the synthesis and secretion of hormones, both cortex and medulla [8,60]. The available literature in this area is mainly concerned with the human adrenal gland. In this aspect, we analyzed the gene expression of macrophages, T and B lymphocytes and NK cells in the adrenal gland of intact adult male rats. At this point, it should be mentioned that a very interesting study by Dolfi et al. [61], characterizing macrophages and other immune cell populations in the adrenal glands of mice, has recently been published. The authors showed that in this species, macrophages constitute a major group (subset) of immune cells, the localization of which depends on sex, mainly on the presence of the adrenal cortical X-zone. In our study, the highest expression levels of macrophage genetic markers were found in the medulla of the gland, while we did not observe significant differences in the expression levels of marker genes of T lymphocytes and NK cells. It is interesting to note that the expression levels of B lymphocyte markers are significantly lower in the medulla of the gland when compared with ZF/R.

At this point in the discussion, we must turn our attention to mast cells in the adrenal gland. They develop from immature (CD34+) hematopoietic precursor cells and settle in organs, forming an important part of the innate and adaptive immune system. Their function is determined by the microenvironment in which they reside [62,63]. This leads to organ-specific heterogeneity of these cells, which causes difficulties in identifying their specific marker genes [44,45]. The previously cited publication on the mouse adrenal gland did not identify these cells [61]. In the rat adrenal gland, they reside mainly in the area under the connective tissue capsule and in the ZG [64]. As the authors suggest, these cells are involved in the regulation of adrenal blood flow. These are very important interstitial cells of the adrenal gland that deserve further detailed study.

Adrenal function is precisely controlled by the nervous system. In the mammalian adrenal glands, the innervation is of both intrinsic and extrinsic types. The extrinsic nerve fibers are conducted mainly by splanchnic nerves, while the intrinsic nerve fibers originate from ganglion cells, which are most abundant in the medulla of the gland and the ZG of the cortex. Numerous neurotransmitters, including catecholamines, as well as biologically active neuropeptides, regulate the function of both the cortex and medulla of the adrenal gland. The regulation of adrenal parenchymal cells occurs both by direct pathways (usually autocrine action mediated by specific receptors) and in an indirect pathway through the regulation of blood flow in the gland [9,65]. In our study, we selected the expression of glial cell marker genes as an indicator of the degree of innervation of the rat adrenal gland. It turned out that the highest expression levels of Cryab and Plp1 genes are found in the adrenal medulla and the lowest in the ZF/R. It is interesting to note that the expression level of these genes in the ZG zone is similar to that observed in M.

Thus, our analysis of the gene expression profiles of both parenchymal and interstitial cells of the adrenal gland of adult male rats under “intact” conditions revealed a number of interesting facts, particularly regarding the interstitial cells involved in creating and maintaining the microenvironment in the gland. Within the gland, the greatest differences in the levels of the marker genes studied concern fibroblasts (highest expression levels in ZG), endothelial cells (different expression levels with three marker genes), VMSCs (lowest expression levels in ZF/R), macrophages (highest expression levels in M), B lymphocytes (highest expression levels in FZ/R) and glial cells (highest expression levels in M). These data suggest that the microenvironment is heterogeneous within the rat adrenal gland. It remains an open question as to what determines the described heterogeneity. It seems reasonable to us to suggest that it depends mainly on the neighboring parenchymal cells of the gland, as suggested by numerous in vitro experiments.

## 4. Materials and Methods

### 4.1. Animals and Experiments

Adult male Wistar rats (12 weeks old; body weight: 120–150 g) were obtained from the Laboratory Animal Breeding Center, Poznan Science and Technology Park of Adam Mickiewicz University Foundation (the SPF category), Poznan, Poland. Animals were maintained in constant, strictly defined atmospheric conditions, i.e., at a temperature of 22 °C +/− 2 °C, air humidity 55–60%, in a daily cycle of 12 h of light (12-h continuum)/12 h of dark (12-h continuum), in a room where the air exchange is at the level of 15 exchanges/h with free access to standard pellets and tap water. All experiments were carried out between 9 and 10 a.m. All possible efforts were made to minimize the number of animals and their suffering.

In brief, the animals (males, n = 6) were subjected to decapitation and their adrenal glands were removed (one sample = two adrenals from one animal). The glands were cleaned of surrounding fat. Under a stereomicroscope, the adrenal glands were decapsulated to separate zona glomerulosa (ZG). Afterward, samples of the zona fasciculata/reticularis (ZF/R) and the medulla (M) were taken from the remaining part of the gland. Isolated adrenal components were submerged in RNAlater and frozen at −80 °C for microarray and qPCR analysis. Unless otherwise stated, all reagents were obtained from Sigma-Aldrich (St. Louis, MO, USA).

### 4.2. RNA Isolation

Total RNA was extracted from samples of entire adrenals using TRI Reagent (Sigma, St. Louis, MO, USA), and next was purified using columns, according to the manufacturer’s instructions (RNeasy Mini Kit, Qiagen, Hilden, Germany). The total mRNA was determined and purity estimated by the method previously described using a NanoDrop spectrophotometer (ThermoScientific, Waltham, MA, USA) and stored at −80 °C for further analysis [66]. The integrity and quality of RNA were checked using the Bioanalyzer 2100 device. The obtained RNA integrity numbers ranged from 8.5–10, with an average of 9.2. Each sample was diluted to a concentration of 100 ng/μL RNA with an OD260/OD280 ratio of 1.8/2.0.

From each RNA sample, 100 ng of RNA was used for microarray experiments. The remaining amount of isolated RNA was used for RT-qPCR studies.

### 4.3. Reverse Transcription and qPCR

To perform the reverse transcription reaction, 100 ng of total RNA was used. Primer 3 software (version 4.1.0) was employed to design the primers (Whitehead Institute for Biomedical Research, Cambridge, MA, USA). The primers were purchased from the Laboratory of DNA Sequencing and Oligonucleotide Synthesis, Institute of Biochemistry and Biophysics, Polish Academy of Sciences, Warsaw, Poland. A list of primers and their sequence can be found in Table 1.

The expression of the investigated genes was measured using quantitative real-time PCR with the CFR96 system from BIO-RAD (Hercules, CA, USA). The reaction was carried out in a 10 μL mixture containing 5 μL of the mix (SsoAdvanced Universal SYBR Green Supermix, BIO-RAD), 1 μL of specific starters, 9 μL of cDNA template and nuclease-free water.

The real-time qPCR reaction utilized standard thermocycling conditions, including a 10-min denaturation step to activate the Taq DNA polymerase. This was followed by a three-step amplification program consisting of denaturation at 95 °C for 3 min, annealing at 95 °C for 10 s and extension at 60 °C for 30 s. All samples were amplified in duplicate. The relative expression of the target genes was calculated using the 2^−ΔΔCt^ quantification method. The analysis was performed in relation to two reference genes—B2m and Hprt.

### 4.4. Microarray RNA Analysis

The microarray study was performed according to the previously described protocol [22]. First, 100 ng of total RNA was subjected to a two-step cDNA synthesis reaction, biotin labeling, and fragmentation using Affymetrix GeneChip^®^ WT Plus Reagent Kit (Affymetrix, Santa Clara, CA, USA). The biotin-labeled cDNA was then hybridized to the Affymetrix^®^ Rat Gene 2.1 ST Array Strip (Affymetrix, Santa Clara, CA, USA), and the array strips were scanned using the Imaging Station from the GeneAtlas System. The Affymetrix GeneAtlas Operating System was used for preliminary analyses with a quality control step.

#### 4.4.1. Microarray Data Analysis

All analyses have been performed by BioConductor software repository (version 3.17) with the relevant Bioconductor libraries by the statistical R programming language (version 4.1.2; R Core Team 2021). The robust multiarray average (RMA) normalization algorithm, which is implemented in the “Affy” library, was applied to normalize, perform background correction and calculate the expression values of the analyzed genes [67]. The gene data table was created by merging the annotated data table from the BioConductor “oligo” package with the normalized data set. Genes with low variance were removed by a variance-based filtering function from “genefilter” library [68]. The “factoextra” library was used to perform principal component analysis (PCA) on a filtered dataset and visualize it in order to demonstrate the total number of up- and down-regulated genes [69]. The “limma” library was utilized to determine the differential expression and statistical assessment through linear models for microarray data, where pairwise comparisons were carried out for ZG vs. ZF/R, ZG vs. M, and ZF/R vs. M. Significantly changed gene expression was determined based on a *p*-value with false discovery rate (FDR) correction < 0.05 and absolute value from fold change ≥2. The results of this selection were presented as a volcano plot, showing the total number of up- and down-regulated genes. Obtained results were visualized using “ggplot2” and “ggprism” libraries [70,71].

#### 4.4.2. Analysis of Gene Clusters Assigned to Different Adrenal Components

A set of mRNAs expression data whose expression was significantly regulated in at least one of the compared pairs were selected for analysis. To determine the optimal number of clusters, we applied repeatedly calculated sum of squared error (SSE) measurement with an increasing number of clusters. The K-means algorithm was used for the clustering of mRNA expression profiles. Clustering was performed on the average expressions from each experimental group using “kmeans”—core R function. For each cluster, the centroids values and core mRNA sets were determined. Core mRNAs were generated by filtering the fitting level of mRNAs expression to centroid values where the mRNA expression profile displayed a high correlation to centroids for a given cluster (correlation > 0.8). The twenty genes with the highest and lowest expression values (if present) for a given group were presented as heatmaps along with hierarchical clustering using the ‘Complexheatmap’ package [72]. Next, the DAVID (Database for Annotation, Visualization and Integrated Discovery) bioinformatics tool was employed to functionally differentially expressed genes (DEGs) assigned to adrenal component-specific clusters [73]. Genes from specific clusters were assigned to relevant Gene Ontology (GO) terms from the GO BP DIRECT database. The *p*-values of the selected GO terms were corrected using the Benjamini–Hochberg correction [74].

Moreover, the gene expression changes were mapped onto the signaling pathways maps from the WikiPathways database [75] using the “rWikiPathways” [76] library and Cytoscape software (version 3.10.0) [77].

The assignment of adrenal component-specific genes to appropriate gene ontology biological terms was also carried out using a cluster profiler library [78]. ENTREZ ids with fold change values of differentially expressed genes were subjected to the analysis. Five top enriched GO BP terms were visualized.

#### 4.4.3. Comparison of Our Results to Previous Studies

The analysis was conducted using a dataset from Nishimoto’s publication on the rat transcriptome profile in ZG and ZF. Data, including fold change and adjusted *p*-value for differentially expressed genes between ZG and ZF, were downloaded from the publisher’s web page (https://academic.oup.com/endo/article/153/4/1755/2423885?login=false#supplementary-data, accessed on 30 April 2023) where they were deposited as Appendix A. To determine similarities and differences in the expression of specific genes obtained in our study in relation to the Nishimoto genes, we used the ggvenn library for analysis with visualization as a Venn diagram [79].

### 4.5. Statistical Analysis

The applied statistical analyses of gene expression are parts of the software used. The real-time qPCR data, on the other hand, are expressed as the median with IQR, and the statistical significance of the differences between the compared groups was estimated using the Mann–Whitney U-test.

## 5. Conclusions

In the absence of precise literature data, the aim of the study was to determine the expression levels of marker genes of both parenchymal and interstitial cells of the adrenal glands of adult male rats under intact conditions, taking into account their localization in the different components of the gland (ZG, ZF/R and M). With regard to adrenal parenchymal cells, the results obtained are as expected. Interesting data were obtained by analyzing the expression level of interstitial cells of the gland, responsible for the formation of a specific microenvironment in the gland. The expression levels of fibroblast marker genes are highest in the ZG and significantly lower in the remaining parts of the gland. The transcriptome profile of marker genes of endothelial cells indicates their considerable heterogeneity throughout the gland. The expression of VSMC marker genes is lowest in ZF/R, and significantly higher in ZG and M. The highest expression levels of macrophage genetic markers are found in the medulla of the gland, while we did not observe significant differences in the expression levels of marker genes of T lymphocytes and NK cells. The highest expression levels of glial cell marker genes are found in the medulla of the gland.

## Figures and Tables

**Figure 1 ijms-24-09159-f001:**
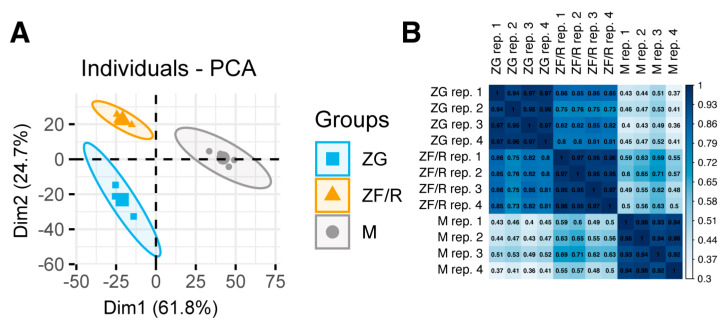
Principal component analysis (PCA) plot showing the filtered distribution of groups of genes expressed in each adrenal zone: glomerulosa (ZG), fasciculate/reticularis (ZF/R) and medulla (M) (**A**). Each dot represents one of the experimental samples assigned to the appropriate experimental group. Pearson’s correlation coefficient analysis array (**B**). The correlation coefficient was calculated for all experimental groups.

**Figure 2 ijms-24-09159-f002:**
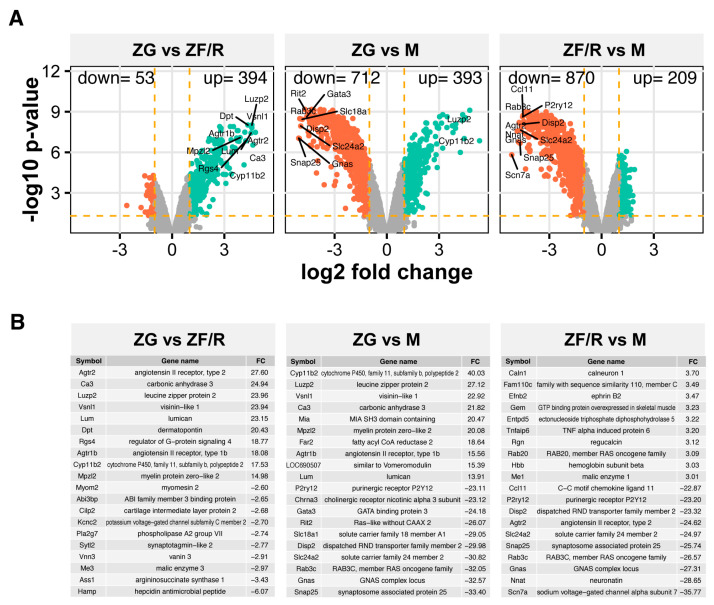
Volcano plots of transcriptome profiling of individual rat adrenal zones (**A**). Comparisons were made between all zones. Each dot in the graph represents the average normalized expression of a single gene, and the orange dashed lines show the cut-off values (absolute fold > 2 and *p* < 0.05 with 20% correction for false discovery rate). Red dots indicate down-regulated genes and green dots indicate up-regulated genes. Several of the most up-regulated genes are denoted by their gene symbols (**A**). Part B is the list of genes with the greatest change in expression—10 up- and 10 down-regulated in each comparison (**B**).

**Figure 3 ijms-24-09159-f003:**
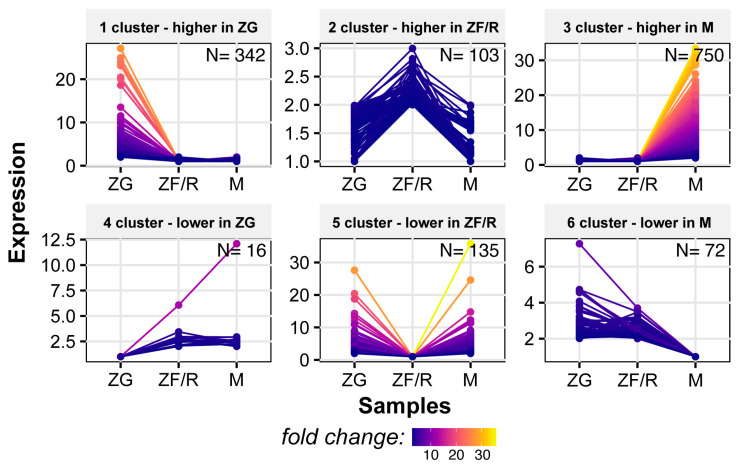
Clustering of differentially expressed genes in every zones of rat adrenal gland. Each dot represents an individual gene where the color corresponds to the level of expression change according to the color scale. The number of genes with variable expression (N) is marked on the top right of the graph.

**Figure 4 ijms-24-09159-f004:**
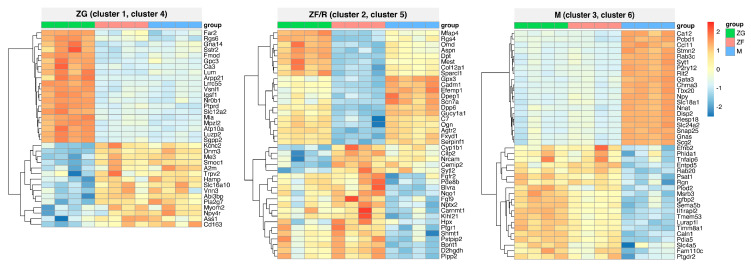
Heatmap showing the expression profiles of the most strongly altered genes from the clusters of Figure 3. The number of genes per cluster shown in the graph is N = 20 or less. Changes in gene expression are shown as color-changing squares, according to the legend. The expression of each gene was normalized to the highest and lowest expression values (row normalization).

**Figure 5 ijms-24-09159-f005:**
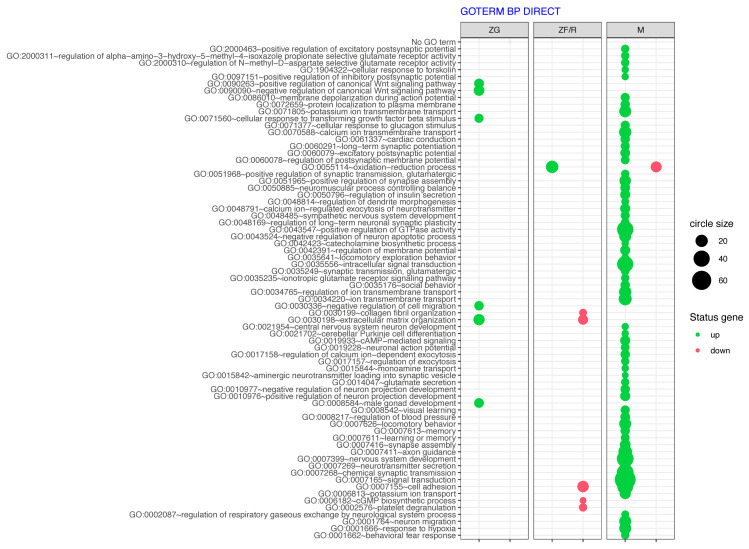
Bubble plot of overrepresented gene set in DAVID GEOTERM BP DIRECT annotation database obtained from all differentially expressed genes in experimental groups, according to clusters of Figure 2. Plots show only GO groups above the established cut-off criterion (*p* corrected < 0.05). The size of each bubble reflects the number of differentially expressed genes assigned to each GO BP term. The transparency of the bubbles displays the *p*-value (more transparent—closer to the cut-off of *p* = 0.05). Genes of ontology groups whose expression increased are shown in green and those whose expression decreased in red.

**Figure 6 ijms-24-09159-f006:**
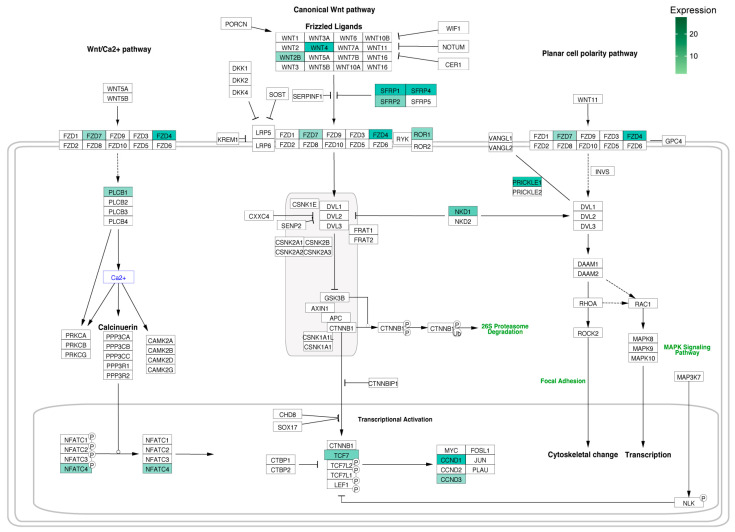
Wnt signaling pathway, including the canonical pathway, the pathway associated with cell polarity determination (Wnt/PCP), and an increase in intracellular Ca^2+^ (Wnt/Ca^2+^), whose genes were significantly regulated in the ZG zone. Each gene involved in the pathway is marked with a separate rectangle. If the expression of a particular gene changed, this was marked with the corresponding color in the diagram, as shown in the legend.

**Figure 7 ijms-24-09159-f007:**
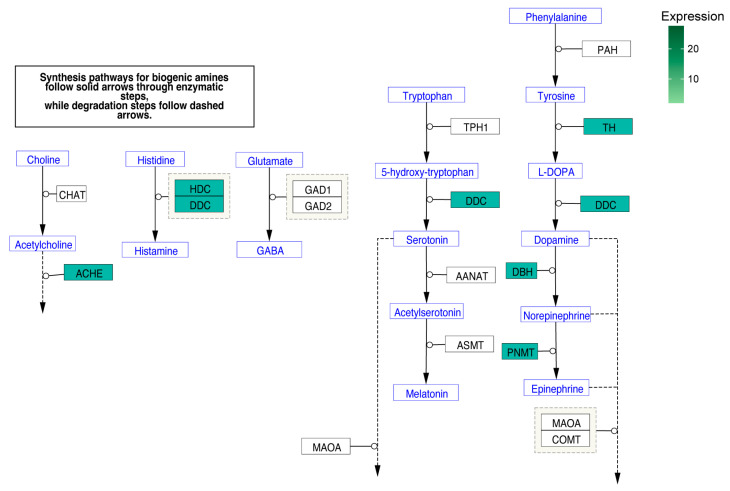
Biogenic amine synthesis signaling pathway, whose genes were characterized by expression significantly regulated in the adrenal medulla. Each gene involved in the pathway is marked with a separate rectangle. If the expression of a particular gene was up-regulated, this was marked with the corresponding color in the diagram, as shown in the legend.

**Figure 8 ijms-24-09159-f008:**
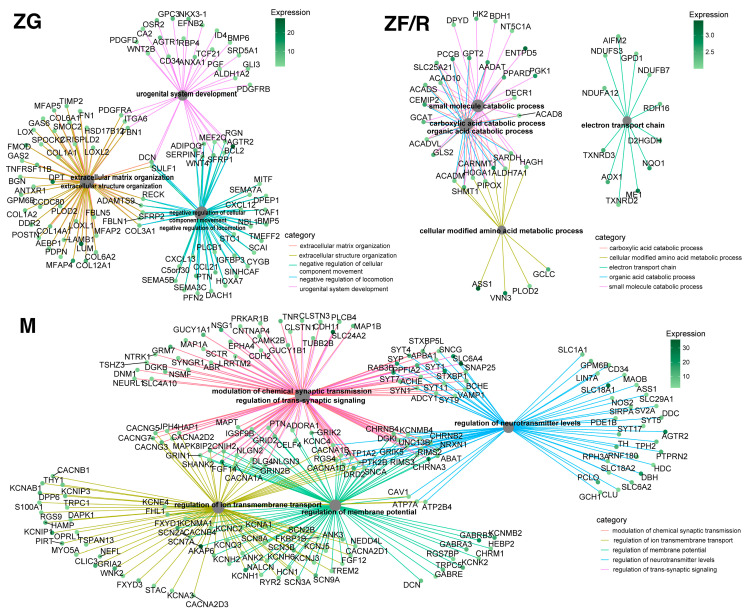
Analysis of the assignment of specific genes to specific ontology groups using the cluster profiler library. Five ontological processes regulated in the most significant way in each of the adrenal zones were selected, and genes with the most altered expression were assigned to them. Changes in gene expression correspond to the legend in the diagram.

**Figure 9 ijms-24-09159-f009:**
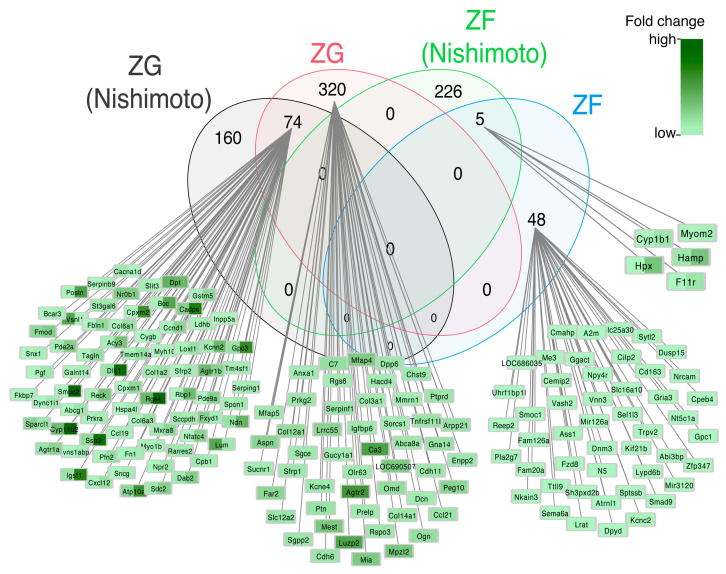
Venn diagram showing total numbers of differentially expressed genes with their common occurrence in the analyzed experimental groups: ZG and ZF. It is a comparison of the results obtained in the present study to those of Nishimoto et al. [21]. The graph shows genes whose expression was statistically significantly altered in the glomerulosa and fasciculata zones of the adrenal cortex (in our study, cells of the fasciculate/reticularis zone were analyzed). The results of the gene expression analysis are shown in circles of different colors: ZG (Nishimoto)—grey, ZF (Nishimoto)—green, ZG (present study)—pink and ZF (present study)—blue. Changes in the expression of individual genes have been highlighted in the appropriate color according to the legend. The expression of common genes was split, and the left part of the rectangle is the result of Nishimoto, and the right part is the result of this paper, as shown on the diagram.

**Figure 10 ijms-24-09159-f010:**
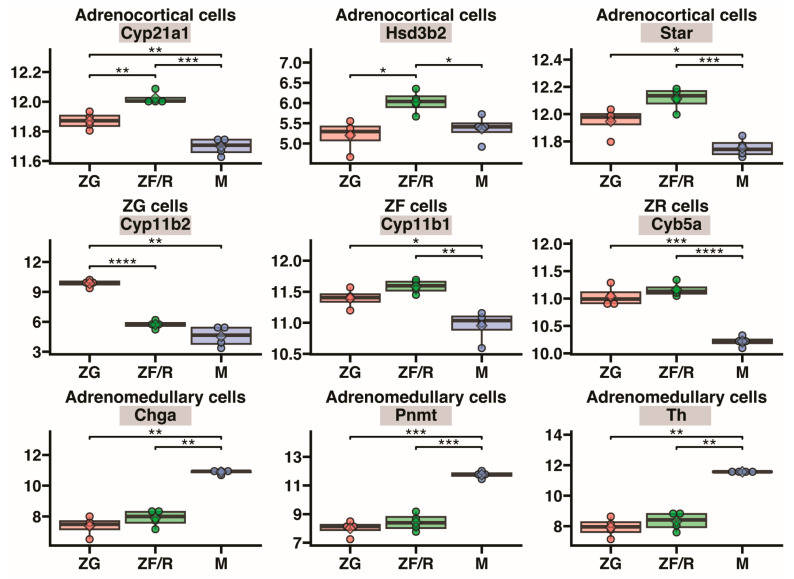
The expression of marker genes in different parenchymal cells of the rat adrenal gland. ZG—zona glomerulosa (red), ZF/R—zona fasciculata/reticularis (green) and M—adrenal medulla (blue). Gene names presented in the diagram are listed in the figure caption. Differences in gene expression in individual layers components of the gland are indicated by asterisks: * *p* < 0.05; ** *p* < 0.02; *** *p* < 0.01; **** *p* < 0.001. The absence of asterisks indicates no statistical significance.

**Figure 11 ijms-24-09159-f011:**
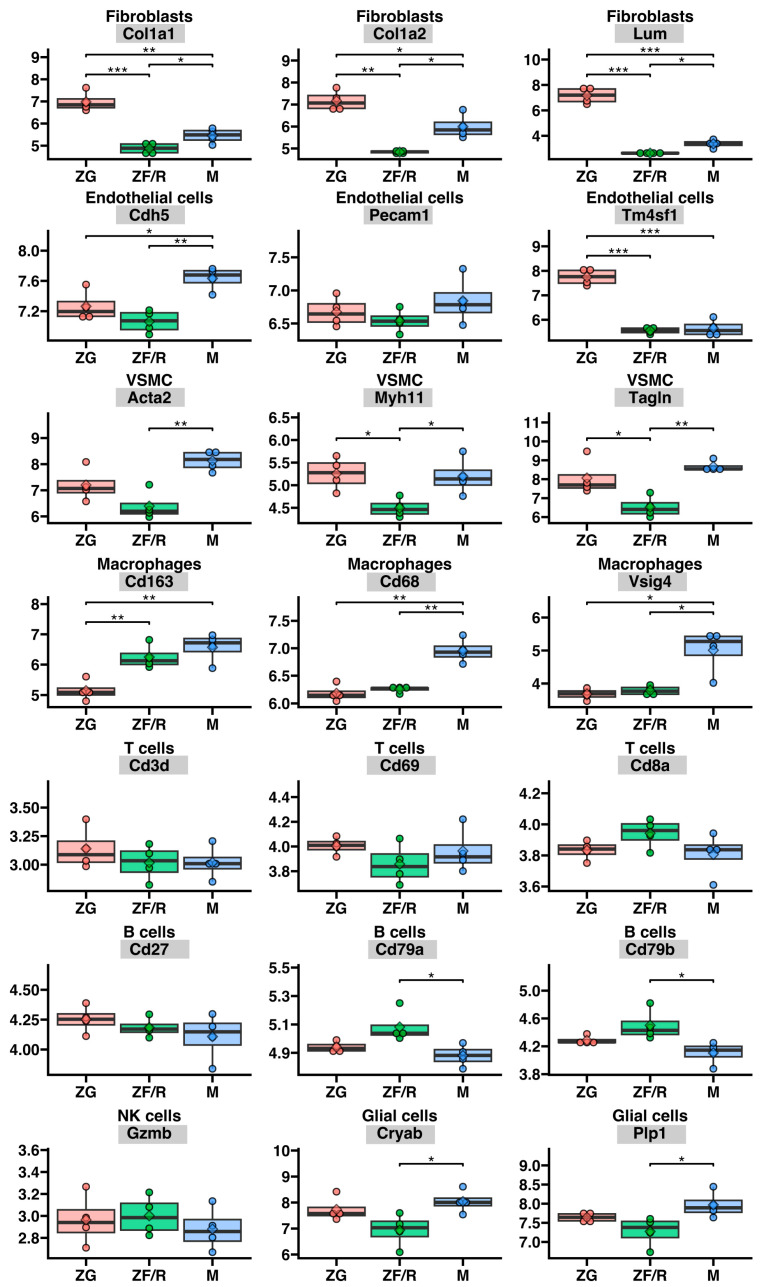
Expression of marker genes of chosen rat adrenal interstitial cells involved in, among others, the formation of the gland’s microenvironment. We studied the markers of the following cells: endothelial cells, fibroblasts, vascular smooth muscle cells (VSMC), macrophages, T and B lymphocytes, natural killer (NK) and glial cells. Gene names presented in the diagram are listed in the figure caption. ZG—zona glomerulosa (red), ZF/R—zona fasciculata/reticularis (green) and M—adrenal medulla (blue). Differences in gene expression in individual components of the gland are indicated by asterisks: * *p* < 0.05; ** *p* < 0.02; *** *p* < 0.01. The absence of asterisks indicates no statistical significance.

**Figure 12 ijms-24-09159-f012:**
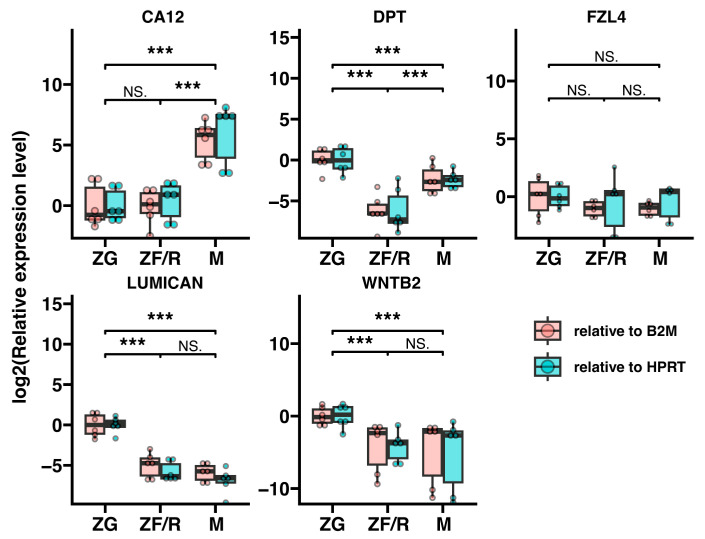
A real-time QPCR study to investigate the expression of selected genes (Ca12, Dpt, Fzl4, Lum and Wnt2b) in different zones of the rat adrenal gland. The bars in the graph represent mean ± SEM (𝑛 = 3), and all samples were amplified in triplicate. The Hprt and B2m genes were used as references to normalize the data. Statistically significant differences between groups were determined as follows: *** *p* < 0.01. NS—not statistically significant.

**Table 1 ijms-24-09159-t001:** QPCR analyses of Ca12—carbonic anhydrase 12, Dpt—dermatopontin, Fzl4—cyclin B, Lum—lumican and Wntb2—Wnt family member 2B. Oligonucleotide sequences for sense (S) and antisense (A) primers are shown. Hprt (hypoxanthine phosphorybosyl transferase) and B2m (beta-2 microglobulin) were the reference genes.

cDNA	GenBank Accession Number	Primer	Primer Sequence (5′–3′)	Position	PCR Product Size (bp)
Ca12	NM_001080756.2	S A	GCACATCGTCCATTATAACT CACTTGTTGGCCTTTATACT	690–709 858–839	169
Dpt	NM_001401366.1	S A	TCCAAAGCCGTTACTTCGAG CATAGTCAGTCATCCGGCAC	377–396 616–597	240
Fzd4	NM_001106818.1	S A	ACAGCTCACAGTCTTTTATCCT TACAAACTGCTCACGACCC	445–466 625–607	181
Lum	NM_031050.2	S A	TGGAGGTCAATAAACTCGAA ACACTCGTACATGTCAGGG	887–906 1029–1011	143
Wnt2b	NM_001191848.1	S A	CGTCTGGGTCTTGCCTGTCT TCCTACTGAGCGCATGATGTCT	112–131 291–270	180
Hprt	NM_012583	S A	CCCCAAAATGGTTAAGGTTG TTCCACTTTCGCTGATGACA	528–547 703–684	176
B2m	NM_012512.2	S A	CTTGCAGAGTTAAACACGTCA CTTGATTACATGTCTCGGTC	316–336 385–366	70

## Data Availability

Data supporting the reported results are included in the Appendix A.

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
