# Peer review of "Transcriptome Profile of the Rat Adrenal Gland: Parenchymal and Interstitial Cells"

_ijms, 2023, doi:10.3390/ijms24119159_

Round 1

Reviewer 1 Report

The manuscript entitled “Transcriptome Profile of the Rat Adrenal Gland: Parenchymal 2 and Interstitial Cells” by Tyczewska et al., addresses very important issue of the genetic description of model system, particularly rats’ adrenal glands. The adrenal cortex originates from two different germ layers mesoderm and neuroectoderm and in adults could be subdivided into different functional zones. Thus gene expression profiling of these tissues is of interest. In their study Authors used Wistar rats to isolate RNA samples and perform gene expression profiling using Illumina Rat Gene 2.1 ST Array. The gens with the specific expression were detected for each studied zone. Overall, the paper is interesting and well written. Data are convincing. There are few minor remarks that would probably improve the paper.

1.       It is unclear how RNA samples from different zones and animals were used for the profiling. Were samples pooled or screened independently? How many samples of each RNA? 6 animals were used for RNA preparation and perhaps only 4 were used for profiling (fig1B). Why only 4?

2.       Previously Nishimoto et al. et al performed similar study using Sprague-Dawley rats and Rat Ref-12 array. It is well documented that SD rats have a higher growth rate and food conversion compared to Wistar rats. Thus it would be very important discussing differences between 2 studied to admit the impact arising from different strain and microarray on gene expression profiling.

3.       Line 56 correct “…..between all types of adrenal gland cells cells maintain…”

4.       Line 74 “Therefore, the aim of our work was to analyze the gene expression profiles in different cell types in the 75 glomerulosa (ZG) and fasciculata-reticularis (ZF/R) zones and in the medulla (M) 76 of the rat adrenal gland.”  In fact Authors did not analyze gene expression profiles in different cell types, they did not perform any cell sorting but analyzed different tissues (mi[ture of cells).

Minor English editing

Author Response

We are grateful for the comments. Thank you for pointing out that our work is important and interesting, and for the opportunity to respond to the thoughtful critique and concerns.

We have highlighted the changes in red in the revised manuscript. Please find below our point-by-point response to your review. We hope that the answers to your questions will be sufficient to publish our results in the International Journal of Molecular Sciences.

  1. It is unclear how RNA samples from different zones and animals were used for the profiling. Were samples pooled or screened independently? How many samples of each RNA? 6 animals were used for RNA preparation and perhaps only 4 were used for profiling (fig1B). Why only 4?

Respond: For the experiment, organs taken from six rats were used. All adrenal glands were used in the study, with adrenal glands taken from one rat constituting 1 sample, from another rat a second sample, and so on. Thus, 6 samples were analyzed in the study (one sample=one rat=two adrenals). The relevant information is included in the section Material and methods of the manuscript (red colour).

  1. Previously Nishimoto et al. et al performed similar study using Sprague-Dawley rats and Rat Ref-12 array. It is well documented that SD rats have a higher growth rate and food conversion compared to Wistar rats. Thus it would be very important discussing differences between 2 studied to admit the impact arising from different strain and microarray on gene expression profiling.

Respond: Thank you for drawing attention to this issue. We are aware of the differences that arise from various rat strains. In fact, previous studies have been conducted regarding the differences among different strains of the species. The findings of this analysis were published in a paper titled "Sex differences in adrenocortical structure and function: XXIV. Comparative morphometric studies on adrenal cortex of intact mature male and female rats of different strains" by Malendowicz, L. K. in 1987. Moreover, it is true that there are certain differences between Wistar and Sprague-Dawley rat strains, which could have also influenced the discrepancies observed in our results compared to those of Nishimoto. The relevant information is included in the manuscript (red colour).

  1. Line 56 correct “…..between all types of adrenal gland cells cells maintain…”

Respond: The mistake has been corrected. One word “cells” have been deleted.

  1. Line 74 “Therefore, the aim of our work was to analyze the gene expression profiles in different cell types in the 75 glomerulosa (ZG) and fasciculata-reticularis (ZF/R) zones and in the medulla (M) 76 of the rat adrenal gland.” In fact Authors did not analyze gene expression profiles in different cell types, they did not perform any cell sorting but analyzed different tissues (mi[ture of cells).

Respond: Thank you for drawing attention. The purpose of the work has been corrected.

Reviewer 2 Report

Comments:

1. Please explain why chose male rats only?

2. The data from ZF/ZR is combined. However, the 2 layers have different enzymes and co-factors requirement. Any significant difference between ZF and ZR in transcriptome profile?

3. It will be a good idea to have a Table summarize the major genes in each layer and their functions in rat adrenal function such as steroidogenesis?

Author Response

We are grateful for the comments. Thank you for the opportunity to respond to the thoughtful critique and concerns.

We have highlighted the changes in red in the revised manuscript. Please find below our point-by-point response to your review. We hope that the answers to your questions will be sufficient to publish our results in the International Journal of Molecular Sciences.

  1. Please explain why chose male rats only?

Respond: In the present study, adrenal gland from males only was used to avoid changes in gene expression due to changes in the estrous cycle.

  1. The data from ZF/ZR is combined. However, the 2 layers have different enzymes and co-factors requirement. Any significant difference between ZF and ZR in transcriptome profile?

Respond: The reticularis zone (ZR) of the rat adrenal gland is not sharply demarcated from the zona fasciculata (ZF). Therefore, in studies describing the fasciculata zone, such samples are commonly referred to as ZF/R. In addition, the cells of the ZR of the rat do not express genes responsible for the synthesis of adrenal androgens - for example, DHEA and DHEAS. Cells of this zone could be collected using "laser-capture microdissection (LCM)," but this requires techniques that can alter gene expression (frozen sections and staining, or paraffin sections and staining). Therefore, we have not performed studies of this type.

  1. It will be a good idea to have a Table summarize the major genes in each layer and their functions in rat adrenal function such as steroidogenesis?

Respond: Yes, thank you for the insightful suggestion. Creating a table would indeed be an excellent idea. In fact, we are currently in the process of planning a paper where we aim to compile the key marker genes for various types of adrenal cell populations, presenting them in a convenient tabular format.

Round 2

Reviewer 2 Report

No more comments